# Diagnosis of Ion-Exchange Resin Depositions in Paraffin Sections Using Corrective Light and Electron Microscopy-NanoSuit Method

**DOI:** 10.3390/diagnostics11071193

**Published:** 2021-06-30

**Authors:** Mako Ooishi, Satoshi Yamada, Toshiya Itoh, Shiori Meguro, Haruna Yagi, Isao Kosugi, Toshihide Iwashita, Kazuya Shinmura, Kiyoshi Misawa, Takahiko Hariyama, Hideya Kawasaki

**Affiliations:** 1Institute for NanoSuit Research, Preeminent Medical Photonics Education & Research Center, Hamamatsu University School of Medicine, Hamamatsu, Shizuoka 431-3192, Japan; A18020@hama-med.ac.jp (M.O.); hariyama@hama-med.ac.jp (T.H.); 2Department of Otolaryngology/Head and Neck Surgery, Hamamatsu University School of Medicine, Hamamatsu, Shizuoka 431-3192, Japan; 41240280@hama-med.ac.jp (S.Y.); kiyoshim@hama-med.ac.jp (K.M.); 3Department of Obstetrics and Gynecology, Hamamatsu University School of Medicine, Hamamatsu, Shizuoka 431-3192, Japan; toshitou@gmail.com; 4Department of Regenerative & Infectious Pathology, Hamamatsu University School of Medicine, Hamamatsu, Shizuoka 431-3192, Japan; megu.s@hama-med.ac.jp (S.M.); hyagi@hama-med.ac.jp (H.Y.); kos180@hama-med.ac.jp (I.K.); toshiiwa@hama-med.ac.jp (T.I.); 5Department of Tumor Pathology, Hamamatsu University School of Medicine, Hamamatsu, Shizuoka 431-3192, Japan; kzshinmu@hama-med.ac.jp

**Keywords:** ion-exchange resin, NanoSuit, corrective light and electron microscopy, energy-dispersive X-ray spectroscopy

## Abstract

Ion-exchange resins are commonly used to treat complications such as hyperkalemia, hyperphosphatemia, and hypercholesterolemia. Gastrointestinal complications may occur as side effects of such treatments. Sodium and calcium polystyrene sulfonate (PS-Ca) are cation-exchange resins comprising an insoluble structure that binds to potassium ions in the digestive tract and exchanges them with sodium and calcium ions, respectively, to promote their elimination. PS crystals are rhomboid, refractive, and basophilic in hematoxylin and eosin staining. To differentiate PS crystals from other ion-exchange resin crystals such as sevelamer and cholestyramine, periodic acid–Schiff, Ziehl–Neelsen, and Congo red staining are usually performed. Here, correlative light and electron microscopy (CLEM)-energy-dispersive X-ray spectroscopy and the NanoSuit method (CENM) was applied to perform a definitive identification of ion-exchange resins. CENM could detect sulfur in PS crystals without destroying the glass slides. Notably, PS retained its ion-exchange ability to bind potassium in paraffin sections. Differential diagnosis of anion-exchange resins, such as sevelamer and cholestyramine, was possible using these characteristics. The phosphorus:carbon ratio was higher in sevelamer than in cholestyramine after soaking paraffin sections in a phosphate solution. Therefore, CENM may be used for the differential pathological diagnosis of ion-exchange resins in paraffin sections.

## 1. Introduction

Ion-exchange resins comprise cross-linked, water-insoluble polymer-carrying, and ionizable functional groups. Cation-exchange resins (CERs) such as sodium and calcium polystyrene sulfonate (PS-Ca) can bind to and trap potassium, whereas anion-exchange resins such as sevelamer and cholestyramine bind to and trap phosphate ions and bile acids. The use of PS to treat hyperkalemia was first described in the 1950s, and the resins are now commonly used to treat hyperkalemia caused by renal failure [1,2]. However, the use of PS is associated with relatively common gastrointestinal (GI) side effects such as nausea and constipation. A possible relationship between GI tract injuries and PS administration, particularly between the development of colonic necrosis and the administration of PS with sorbitol, has been suggested [3,4]. Sevelamer (PHOSBLOCK^®^) is a calcium-free phosphate binder that does not create a positive calcium balance caused by calcium-based binders that are associated with increased mortality in patients with chronic kidney disease (CKD) [5]. Cholestyramine (Questran) is an anion-exchange resin used to treat hypercholesterolemia and diarrhea triggered by Crohn’s disease, ileal resection, vagotomy, or diabetic neuropathy. It prevents the reabsorption of bile acids, thereby allowing hepatic cholesterol to be converted to bile acids which lowers the plasma levels of low-density lipoprotein cholesterol [6]. Crystals associated with this medication have been linked to erosion, ulceration, and acute inflammation that most often occur in the colon.

Pathologists often detect GI erosion or necrosis with foreign body depositions that affect diagnosis. In particular, when the history of ion-exchange resin administration is not documented in any pathology requisition, awareness of the characteristic morphology of ion-exchange resins is crucial to avoid diagnostic pitfalls of its mimics. PS crystals with a unique rhomboid shape and narrow and rectangular “fish scales” were identified in a tissue specimen via basophilic staining using hematoxylin and eosin (H&E) stains [7]. Swanson et al. [8] described sevelamer crystals for the first time in 2013 as rusty yellow-brownish crystals with a characteristic “fish scale” pattern. They show a variable rusty brown color in eosinophils stained via H&E and are embedded inside an eroded or ulcerated GI tract mucosa in patients with CKD. Cholestyramine crystals lack internal “fish scales”, and are bright orange when stained via H&E. The color characteristics of these crystals have also been analyzed following periodic acid–Schiff (PAS), Ziehl–Neelsen (ZN), and Congo red staining of tissues to differentiate them from sevelamer or cholestyramine. This aids in the diagnosis of deposition with clinical drug history [9]. PAS staining displays magenta for CER crystals, violet for sevelamer, and gray or hot pink for cholestyramine [10]. ZN staining displays magenta for CER crystals but does not stain cholestyramine. Congo red staining does not stain CER crystals; however, it stains cholestyramine in a bright vermilion color [9].

We recently developed a new correlative light and electron microscopy (CLEM) method using NanoSuit on paraffin sections, which enables not only CLEM but also energy-dispersive X-ray spectroscopy (EDS) analysis [11]. NanoSuit is a natural polymerized extracellular substance present on the outer surface of animals, and is formed via electron beam or plasma irradiation, resulting in the formation of a nanoscale layer; the layer enables small animals to remain alive under scanning electron microscopy (SEM) observation conditions [12]. The NanoSuit-CLEM method requires only a few minutes to prepare samples without destruction of the specimens. Shinmura et al. used this method on lanthanum depositions in paraffin sections of the GI tract for a definitive diagnosis [13]. To the best of our knowledge, there are no reports on the elemental analysis of ion-exchange resins in paraffin sections for performing a definitive diagnosis. Here, we presented the utility of the NanoSuit-CLEM method in analyzing ion-exchange resins associated with various diseases.

## 2. Materials and Methods

### 2.1. Patient Selection and Sample Collection

Nine patients diagnosed with PS-Ca (kalimate) deposition at the Hamamatsu University Hospital and Chutoen General Medical Center in Japan were selected. H&E-stained, formalin-fixed, and paraffin-embedded slides of the GI tissue containing unique rhomboid-shaped crystals determined via basophilic staining, which were tentatively identified as PS-Ca (kalimate) depositions in routine pathological diagnosis, were used in this study. All patients had a history of calcium PS treatment (Table 1). The study was conducted in accordance with the guidelines of the Declaration of Helsinki and approved by the Ethics Committee for Clinical Research of Hamamatsu University School of Medicine (reference number: 18-074) and Chutoen General Medical Center (No. 147). Informed consent was obtained from all participants.

### 2.2. NanoSuit-CLEM Method Combined with SEM-EDS

The NanoSuit-CLEM method enabled a versatile study of the same structures by CLEM to observe tissue and pathogens with several analytical methods in an SEM without destruction of paraffin sections on slide glass [11,13]. First, the cover glass was removed using xylene, and the section on the slide was rehydrated using a drop of surface shield enhancer (SSE) solution (NanoSuit solution). The stock solution of SSE consisted of sucrose, fructose, and sodium chloride dissolved in distilled water, to which citric acid and sodium glutamate (pH 7.4) were added [14]. The resulting aqueous solution was mixed with glycerin at a ratio of 1:2. SSE was diluted 20-fold in ethanol. The sections were subsequently spin-coated via centrifugation at 6200 rpm for 15 s to remove excess SSE solution [11]. The procedure for CLEM-EDS and NanoSuit method (CENM) is as follows: The specimens were directly added to the SEM system to form the NanoSuit following irradiation with an electron beam. Elemental analysis was performed using an SEM system (TM4000Plus; HITACHI, Tokyo, Japan; accelerating voltage: 15 kV) equipped with an EDS (X-stream-2; Oxford Instruments, Oxford, UK). The AZtecOne software (Oxford Instruments) was used for the analysis of EDS data and EDS mapping. The weight concentrations of all detected elements were quantified and expressed as weight percentage (wt%), which indicates the relative concentration of an element at the analyzed point. After ultrastructural observation, the specimens were re-stained with H&E, and the cover glass was mounted using diaphane (Malinol, Muto Pure Chemicals, Tokyo, Japan) for storage [11].

### 2.3. Preparation of Cholestyramine and Sevelamer Paraffin Sections

Suspensions of cholestyramine (SANOFI K.K., Tokyo, Japan) and sevelamer (Kyowa Kirin Co., Ltd., Tokyo, Japan) were dissolved in 2% agar in saline. Solid agar containing the drugs was fixed in 10% formalin for 24 h. Normal tissue processing for paraffin sections (ethanol and xylene infiltration, and embedding in paraffin wax) was performed [9]. The sections were then prepared for normal histological analysis and examined via H&E, PAS, ZN, and Congo red staining [7,9].

### 2.4. Preparation of KCl and Phosphate Solution

KCl (Katayama Chemical Industries Co., Ltd., Osaka, Japan) was used to prepare a 100 mEq KCl solution dissolved in water. Ammonium dihydrogenphosphate (Kanto Chemical Co., Inc., Tokyo, Japan) was used as a phosphate-containing solution. Monobasic ammonium phosphate (460 mg) was mixed with purified water (200 mL) to obtain a 20 mM phosphate solution at pH 6 [15]. The ion-exchanged resin-deposited paraffin sections were soaked for 6 h at 37 °C.

## 3. Results

### 3.1. Morphological and Staining Characteristics of Ion-Exchange Resin Depositions on Paraffin Sections

We prepared samples from nine patients who were found to contain PS depositions based on the assessment of pathologists. We reconfirmed the unique rhomboid shapes based on basophilic color (purple) in H&E staining (Figure 1a) in eight out of nine samples. To differentiate the PS crystals from cholestyramine and sevelamer, we performed PAS, ZN, and Congo red staining. All eight samples that were suspected to contain PS showed the same staining patterns (purple in H&E staining, Figure 1a; deep magenta in PAS staining, Figure 1b; deep magenta in ZN staining, Figure 1c; purple in Congo red staining, Figure 1d). Sevelamer crystals were orange-red in H&E staining (Figure 1e), purple in PAS staining (Figure 1f), pink in ZN staining (Figure 1g), and dark purple in Congo red staining (Figure 1h). Cholestyramine crystals were orange-red in H&E staining (Figure 1i), light purple in PAS staining (Figure 1j), light pink or unstained in ZN staining (Figure 1k), and purple in Congo red staining (Figure 1l). Sevelamer and cholestyramine crystals showed similar staining patterns in our experiments.

### 3.2. Identification of Sulfur in Calcium PS Crystals via the NanoSuit EDS Method

The CENM methods were performed on the PS deposition-suspected paraffin sections. First, a representative patient sample (patient 1) was analyzed. The deposited crystals showed unique rhomboid shapes with a basophilic color (purple) (Figure 2a). Three-dimensional calcium PS crystals were observed in the correlative paraffin sections via desktop SEM in a back-scattered mode (Figure 1b). The molecular formula of PS-Ca (kalimate) is C_8_H_7_CaO_3_S^+^, and it is composed of calcium and 2-ethenylbenzosulfonate (Figure 2c) [16]. Elemental sulfur (Figure 2d), carbon (Figure 2e), and oxygen (Figure 2f), but not potassium (Figure 2g), were identified in exactly the same PS crystal positions on the EDS map. Among the elements analyzed, sulfur was the most effective marker for distinguishing the calcium PS crystals (Figure 2d–g). The spectral analysis confirmed the sulfur signal (Figure 2h) in PS crystals; however, fewer signals were detected in the background (Figure 2i).

Eight out of nine patients showed a higher sulfur component than that of the background (Figure 2j and Appendix A). Patient 5 showed no difference in sulfur content (wt%) between the crystal and background areas (Figure 2j and Appendix A). Strong back-scattered electron (BSE) signals (Appendix A) were observed in the BSE mode, and lanthanum (Appendix A) and phosphorus (Appendix A) elements were identified in the crystal area in the paraffin sections via CENM. H&E staining at the pre-EDS analysis showed almost no change except for partial beam marks after EDS analysis (Appendix A).

### 3.3. Ion-Binding Capacity of Ion-Exchange Resins in Paraffin Sections

We investigated whether the ion-binding capacity was retained by the ion-exchange resins in the paraffin sections. First, the sections with ion-exchange resins in paraffin sections were soaked in a KCl solution (100 mEq) for 6 h and washed with water three times. EDS analysis showed the elemental composition of the PS crystals on the map (Figure 3a), spectral data (Figure 3b), and weight (%) data (Figure 3c). Sevelamer and cholestyramine showed a low binding capacity for potassium ions (Figure 3d–i). The binding capacities of PS-Ca and sevelamer or cholestyramine were significantly different (*p* < 0.01) (Figure 3j).

Next, sections with ion-exchange resins of paraffin sections were soaked in phosphate solution (20 mM) for 6 h and washed with water thrice. EDS did not show elemental phosphorus with PS crystals in the EDS map (Figure 4a), spectral data (Figure 4b), or weight (%) data (Figure 4c) in the PS-Ca crystals. In contrast, sevelamer and cholestyramine showed binding capacity for phosphorus ions (Figure 4d–i). The binding capacity of PS-Ca and sevelamer/cholestyramine for phosphorus was significantly different (*p* < 0.01) (Figure 4j).

### 3.4. The Possibility of Differential Diagnosis between Sevelamer and Cholestyramine Deposition in Paraffin Sections

The binding capacities of sevelamer and cholestyramine for phosphorus were compared using random sampling from different drug crystals using point EDS analysis. The proportion of phosphorus (weight %) per carbon element (weight %) was significantly different (*p* < 0.01) (Figure 5a, Appendix A). A receiver operating characteristic (ROC) curve was generated to evaluate the diagnostic cut-off value. ROC curve analysis showed that 0.1435 (phosphorus (weight %)/carbon (weight %)) was the optimal cut-off value for differential diagnosis of sevelamer and cholestyramine (area under curve 0.995, 95% CI: 0.982–1.000; *p* < 0.0001) (Figure 5b and Appendix A).

## 4. Discussion

Pathologists often detect ulcerative lesions in the GI tract with foreign body depositions. The normal and atypical symptoms associated with drug resins should be considered so that based on an early feedback, a clinician terminates the treatment, thereby preventing the development of more severe adverse effects. Recognition and timely diagnosis can be lifesaving. The pathophysiology of drug crystal-induced intestinal necrosis is not well understood. Kim et al. found that crystals of sevelamer, PS, and cholestyramine induce intestinal epithelial cell barrier dysfunction and trigger the formation of neutrophil extracellular traps (NETs) and monocyte/macrophage extracellular traps. In addition, they detected NET formation in a patient with CKD showing sevelamer crystal-related intestinal necrosis [17].

Although morphological analysis is the gold standard for identifying resins, the color, location, and pattern of fish scales may vary, preventing accurate identification. According to Gonzalez et al., it is critical to identify resin crystals correctly because their clinical consequences and treatments vary. However, in an online poll, only 76% of pathologists were reported to correctly identify sevelamer, sodium polystyrene sulfonate, and bile acid sequestrants by H&E staining [18]. It is beneficial to consult with the clinician regarding the drug administered to a patient [19]. We applied the EDS method to diagnose ion-resins for the first time using SEM with NanoSuit to help with accurate diagnosis.

The benefits of the NanoSuit method include a charge-reducing effect without using heavy metals, and it is easy and quick to perform without requiring specialized equipment such as a draft chamber or sputtering equipment. The use of the NanoSuit method in medical applications has gradually increased [13,20,21,22]. The NanoSuit membrane is very thin and does not interfere with the analysis of each element. NanoSuit has been shown to protect entire H&E-stained paraffin sections against drying, preserve elements and small molecules for EDS analysis, and enable re-staining with H&E [11,13]. However, the localized electrical beam marks remained after the EDS analysis in the present study because of the use of an electric beam at the maximum level (Appendix A). The residual beam marks enable detection of the observed area after EDS analysis using a light microscope. However, it is necessary to develop a more conductive NanoSuit solution in the future. 

Sulfur present in crystals on paraffin sections may serve as a distinctive diagnostic marker detected via EDS to differentiate sevelamer and cholestyramine (Figure 2 and Figure 3, and Appendix A). We described the use of an alternative definitive diagnostic method (CENM) for PS instead of/in addition to the use of PAS, ZN, and Congo red staining in this study. CENM is easy and time saving. Table 2 shows a comparison of conventional staining methods and CENM. Notably, the binding capacity of PS-Ca to potassium is still retained after the preparation of tissue sections (fixation, dehydration, clearing, wax infiltration, deparaffinization). These ion-binding characteristics may also be used for diagnosis. The chemical formulas of sevelamer and cholestyramine are C_6_H_12_ClNO and C_21_H_30_ClN, respectively. Both the drugs possess similar chemical formulas, except for the oxygen content. CENM could not differentiate between the two drugs even though sevelamer contains oxygen (O) (Figure 3d–i). It is well known that sevelamer is a calcium-free phosphate-binder, and cholestyramine functions by binding to bile acids. However, the results of the present (Figure 4) and a previous [21] study showed that sevelamer and cholestyramine have phosphate-binding capacities. Oda et al. reported that the phosphate-binding capacity ranges as follows: sevelamer hydrochloride (614 mg/g) > lanthanum carbonate (557 mg/g) > calcium carbonate (535 mg/g) > colestimide (256 mg/g) > cholestyramine (98 mg/g) > colestipol (93 mg/g), indicating that sevelamer hydrochloride is the most efficient phosphate-binder [23]. We succeeded in using their phosphate-binding capacity for differential diagnosis (Figure 4 and Figure 5). We could not find any pathological cases of sevelamer and cholestyramine depositions in this study. The agar model was used to test our hypothesis. The data showed that this technology can be used in actual pathological diagnosis after soaking sections in phosphate buffer. In the actual pathological diagnosis, the reaction may differ slightly for each sample. It is, therefore, necessary to compare it with a positive control sample of sevelamer or cholestyramine.

## 5. Conclusions

The NanoSuit method for EDS can be used for the analysis of ion-exchange resins in paraffin sections. This method will become a routine diagnostic tool in the future as the development of desktop SEM with EDS for CLEM becomes cheaper.

## Figures and Tables

**Figure 1 diagnostics-11-01193-f001:**
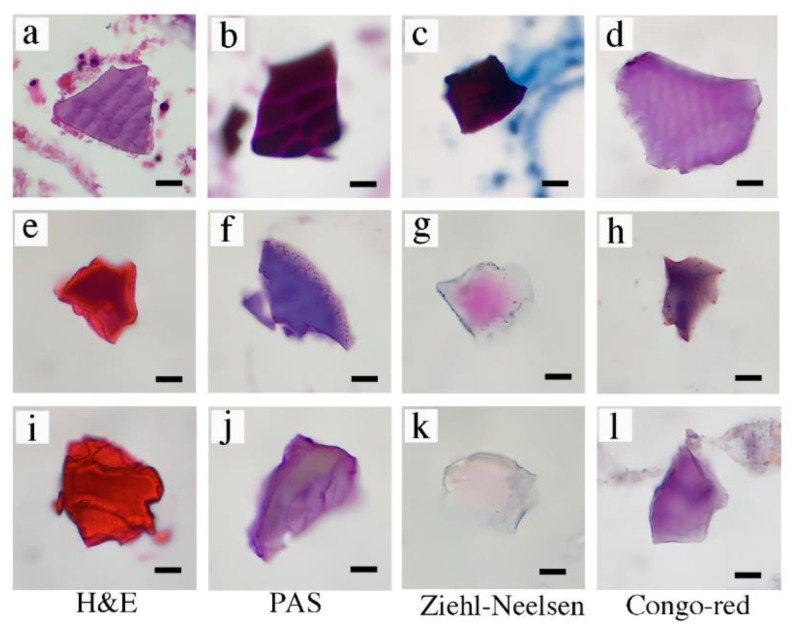
Morphological and staining characteristics of crystals in histopathological sections. (**a**–**d**) Calcium polystyrene sulfonate. (**e**–**h**) Sevelamer. (**i**–**l**) Cholestyramine. H&E (**a**,**e**,**i**), PAS (**b**,**f**,**j**), Ziehl–Neelsen (**c**,**g**,**k**), Congo red (**d**,**h**,**l**) staining. Bar represents 20 μm. H&E, hematoxylin and eosin; PAS, periodic acid–Schiff.

**Figure 2 diagnostics-11-01193-f002:**
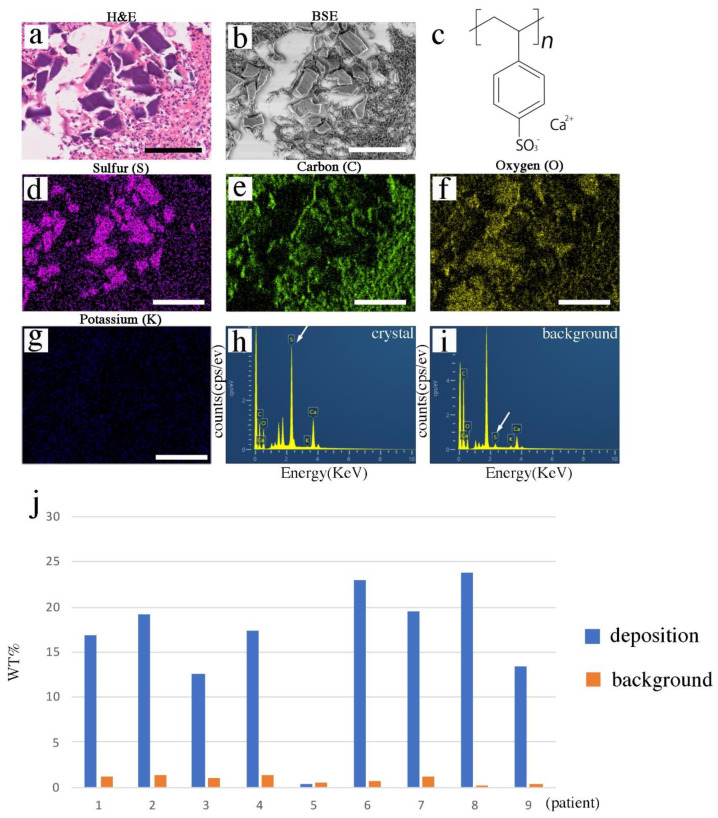
EDS analysis of paraffin sections. (**a**) H&E staining. (**b**) SEM image (BSE mode). (**c**) Chemical formula of the calcium PS. (**d**) Sulfur (S) deposition (purple). (**e**) Carbon (C) (green). (**f**) Oxygen (O) (yellow). (**g**) Potassium (K) (blue). (**h**,**i**) Spectrum analysis of the SEM-EDS using the NanoSuit-CLEM method. Analysis of the crystal (**h**), and (**i**) background. White arrow shows the peak signals of sulfur. (**j**) Comparison of weight percentage (wt%) of sulfur in crystal and background areas among patients. EDS, energy-dispersive X-ray spectroscopy; H&E, hematoxylin and eosin; SEM, scanning electron microscopy; BSE, back-scattered electron; PS, polystyrene sulfonate; CLEM, correlative light and electron microscopy. Bar represents 100 μm.

**Figure 3 diagnostics-11-01193-f003:**
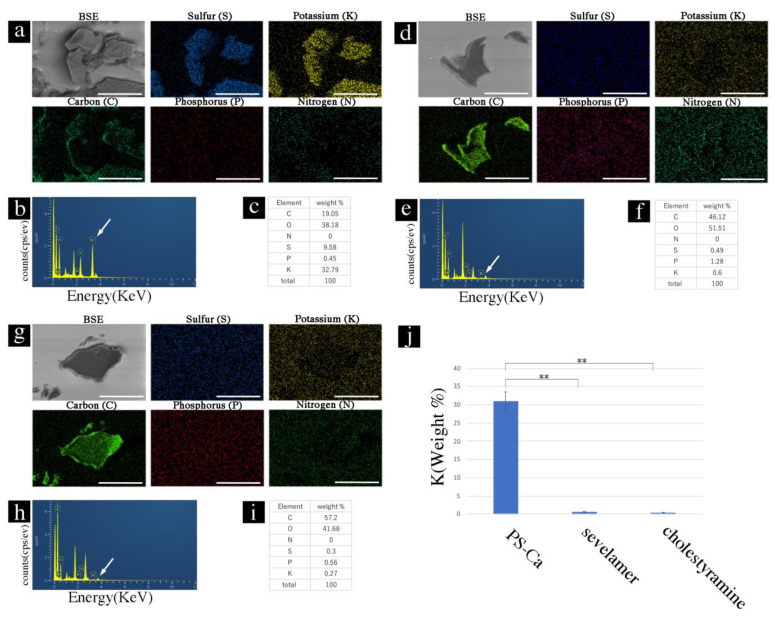
Potassium-binding capacity of ion-exchange resins in paraffin sections. (**a**) PS-Ca, SEM image, and EDS map. (**b**) Spectrum analysis of SEM-EDS using the NanoSuit-CLEM method. White arrow shows the peak signals of potassium. (**c**) Weight % data. (**d**) SEM image and EDS map of sevelamer. (**e**) Spectrum analysis of SEM-EDS using the NanoSuit-CLEM method. White arrow shows the peak signals of potassium. (**f**) Weight % data. (**g**) SEM image and EDS map. (**h**) Spectrum analysis of SEM-EDS using the NanoSuit-CLEM method. White arrow shows the peak signals of potassium. (**i**) Weight % data (**j**) Comparison of the K-binding ratio (weight %) among PS-Ca, sevelamer, and cholestyramine. ** indicates *p* < 0.01. EDS, energy-dispersive X-ray spectroscopy; SEM, scanning electron microscopy; CLEM, correlative light and electron microscopy. Bar represents 50 μm.

**Figure 4 diagnostics-11-01193-f004:**
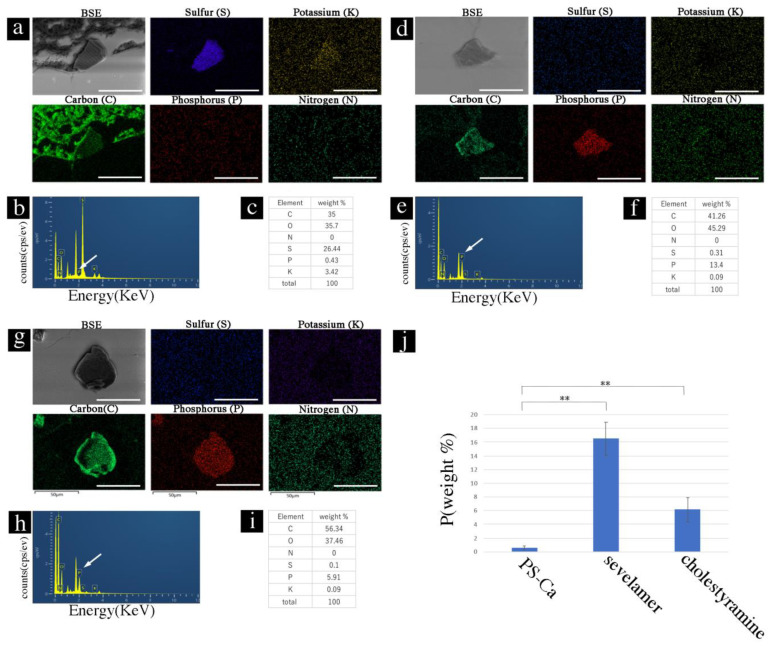
Phosphate-binding capacity of ion-exchange resins in paraffin sections. (**a**) PS-Ca, SEM image, and EDS map. (**b**) Spectrum analysis of the SEM-EDS using the NanoSuit-CLEM method. White arrow shows the peak signals of phosphorus. (**c**) Weight % data. (**d**) SEM image and EDS map of sevelamer. (**e**) Spectrum analysis of the SEM-EDS using the NanoSuit-CLEM method. White arrow shows the peak signals of phosphorus. (**f**) Weight % data. (**g**) SEM image and EDS map. (**h**) Spectrum analysis of the SEM-EDS using the NanoSuit-CLEM method. White arrow shows the peak signals of phosphorus. (**i**) Weight % data. (**j**) Comparison of the P-binding ratio (weight %) among PS-Ca (kalimate), sevelamer, and cholestyramine. ** indicates *p* < 0.01. EDS, energy-dispersive X-ray spectroscopy; SEM, scanning electron microscopy; CLEM, correlative light and electron microscopy. Bar represents 50 μm.

**Figure 5 diagnostics-11-01193-f005:**
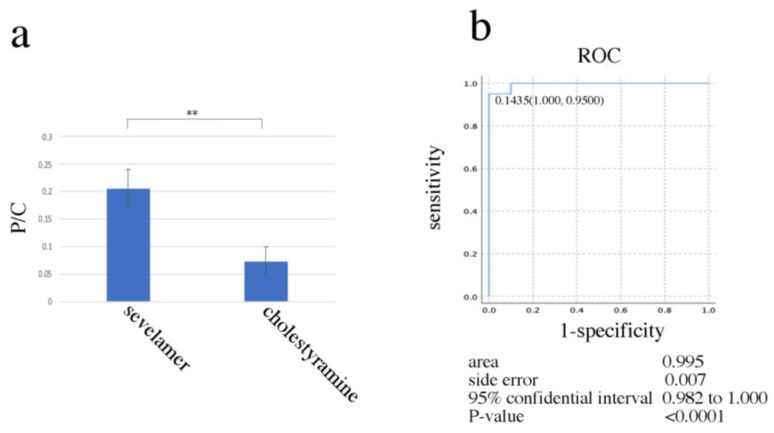
The NanoSuit-EDS method for differentiation between sevelamer and cholestyramine using random sampling. (**a**) Comparison of the P/C ratio between sevelamer and cholestyramine (n = 40). ** indicates *p* < 0.01. (**b**) Accuracy of visual diagnosis was determined based on the area under the ROC curve (n = 40). P/C, phosphorus/carbon; ROC, receiver operating characteristic.

**Table 1 diagnostics-11-01193-t001:** Patients’ clinical information.

No.	Patient Age	Sex	Type of Specimen	Site	Clinical History Provided	Associated Tissue Reaction	Background Renal Disease/History of Drug Intake
1	69	M	Biopsy	Sigmoid colon, rectum	Colonoscopy, longitudinal ulceration	Severe colitis	CKD with dialysis/calcium polystyrene sulfonate 15 days
2	80	F	Surgical specimen	Sigmoid colon	Perforation	Ulcer	CKD/calcium polystyrene sulfonate 5 months
3	83	M	Biopsy	Sigmoid colon, rectum	Colonoscopy, gastrointestinal bleeding	Erosive colitis	Ulcerative colitis/calcium polystyrene sulfonate 25 days
4	78	M	Biopsy	Transverse colon	Colonoscopy, polyp	Erosive colitis, inflammatory polyp	CKD/calcium polystyrene sulfonate 43 days
5	78	F	Biopsy	Stomach	Gastroscopy, cancer or amyloid deposition suspected	Gastritis	CKD/calcium polystyrene sulfonate 7 days, lanthnum carbonate 6 months
6	76	F	Surgical specimen	Sigmoid colon	Perforation	Ulcer	CKD/calcium polystyrene sulfonate 6 days
7	45	M	Biopsy	Transverse colon	Colonoscopy	White nodule	CKD/calcium polystyrene sulfonate unknown period, lanthunm carbonate unknown period
8	50	M	Surgical specimen	Descending colon	Surgery, ulcerative colitis	Severe colitis	Ulcerative colitis/calcium polystyrene sulfonate 18 days
9	69	F	Biopsy	Ascending colon	Colonoscopy, ulcerative Colitis	Ulcer	CKD/calcium polystyrene sulfonate unknown period

**Table 2 diagnostics-11-01193-t002:** Comparison of conventional staining methods and CENM.

	Conventional Staining Method (PAS, ZN, Congo-Red Staining)	CENM
Time	30–60 min	Approximately 5 min
Number of sections	Multiple	Single
Procedure	Multiple steps	A few steps
CLEM	No	Yes
Examination of the ion-exchange capacity	No	Yes

CLEM, correlative light and electron microscopy. CENM, CLEM-EDS and NanoSuit method.

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
