# Peer review of "Diagnosis of Ion-Exchange Resin Depositions in Paraffin Sections Using Corrective Light and Electron Microscopy-NanoSuit Method"

_diagnostics, 2021, doi:10.3390/diagnostics11071193_

Round 1

Reviewer 1 Report

Comments to the Authors The authors investigated the possible effectiveness of corrective light and electron microscopy-NanoSuit method (CENM) in the diagnosis of ion-exchange resin deposition. General comments: I think that this manuscript is well written; however, the authors should describe the usefulness of CENM, especially from the clinical point of view. What kind of advantage does the CENM have in comparison with the conventional method of staining such as PAS, Ziehl-Neelsen, and Congo red staining? Comparison between these methods is recommended.

Author Response

Reviewer 1

Comments to the Authors The authors investigated the possible effectiveness of corrective light and electron microscopy-NanoSuit method (CENM) in the diagnosis of ion-exchange resin deposition. General comments: I think that this manuscript is well written; however, the authors should describe the usefulness of CENM, especially from the clinical point of view. What kind of advantage does the CENM have in comparison with the conventional method of staining such as PAS, Ziehl-Neelsen, and Congo red staining? Comparison between these methods is recommended.

Response:

We would like to thank the reviewer for positive comments and valuable suggestions.

The advantages of the CENM over conventional staining methods are as follows: 

  1. The CENM procedure is easy and takes approximately 5 minutes.
  2. Only one section is required for CENM.
  3. Troublesome procedures, such as staining and multiple washings, in PAS, Ziehl–Neelsen, and Congo red staining are not involved in CENM.
  4. Same resin crystals can be visualized by light microscopy and SEM.
  5. CENM can be used for examining the ion-exchange capacity of ion-exchange resins.

The advantages mentioned in points 1, 2, and 3 above can reduce the burden on technicians and help save time and those mentioned in points 4 and 5 facilitate the examination of new characteristics, which cannot be investigated using the conventional staining methods.

These advantages have been listed in Table 2 (presented below) that has been added to the revised manuscript. Moreover, we have added the following sentences on page 10 (lines 302–303): “CENM is easy and time saving. Table 2 shows a comparison of conventional staining methods and CENM.”

Table 2: Comparison of conventional staining methods and CENM(Please see the attachment)

Reviewer 2 Report

In the present paper, authors described the utility of  NanoSuit-CLEM method for identification of ion-exchange resins. They conclude that CENM may be used for the differential pathological diagnosis of ion-exchange resins in paraffin sections.

Manuscript is overall well written and data are clearly presented; however discussion section should be implemented with few sentences on the clinical utility of differentiating ion-exchange resins in paraffin sections.  In this regard, the pathological differential diagnosis of such resins is not a common practice in pathology laboratories; therefore, authors should better clarify what are the clinical and pathological conditions that require this differentiation.

Author Response

Reviewer 2

In the present paper, authors described the utility of NanoSuit-CLEM method for identification of ion-exchange resins. They conclude that CENM may be used for the differential pathological diagnosis of ion-exchange resins in paraffin section.

Manuscript is overall well written and data are clearly presented; however discussion section should be implemented with few sentences on the clinical utility of differentiating ion-exchange resins in paraffin sections.  In this regard, the pathological differential diagnosis of such resins is not a common practice in pathology laboratories; therefore, authors should better clarify what are the clinical and pathological conditions that require this differentiation.

Response:

We would like to thank the reviewer for positive comments and valuable suggestions. As mentioned by the reviewer, the pathological differential diagnosis of such resins is not a common practice in pathology laboratories. However, clinicians must be mindful of the normal and atypical presentations of drug resins, and therefore, an early feedback to them will result in the medication being stopped and more severe adverse effects being avoided. We have described this in the newly added sentences (page 9, lines 279–282) and have cited a relevant reference: “According to Gonzalez et al., it is critical to identify resin crystals correctly because their clinical consequences and treatments vary. However, in an online poll, only 76% of pathologists were reported to correctly identify sevelamer, sodium polystyrene sulfonate, and bile acid sequestrants by H&E staining [17].”
